# Vision at A Glance: Interplay between Fine and Coarse Information Processing Pathways

## Abstract

Object recognition is often viewed as a feedforward, bottom-up process in machine learning, but in real neural systems, object recognition is a complicated process which involves the interplay between two signal pathways. One is the parvocellular pathway (P-pathway), which is slow and extracts fine features of objects; the other is the magnocellular pathway (M-pathway), which is fast and extracts coarse features of objects. It has been suggested that the interplay between the two pathways endows the neural system with the capacity of processing visual information rapidly, adaptively, and robustly. However, the underlying computational mechanism remains largely unknown. In this study, we build a two-pathway model to elucidate the computational properties associated with the interactions between two visual pathways. The model consists of two convolution neural networks: one mimics the P-pathway, referred to as FineNet, which is deep, has small-size kernels, and receives detailed visual inputs; the other mimics the M-pathway, referred to as CoarseNet, which is shallow, has large-size kernels, and receives blurred visual inputs. The two pathways interact with each other to facilitate information processing. Specifically, we show that CoarseNet can learn from FineNet through imitation to improve its performance considerably, and that through feedback from CoarseNet, the performnace of FineNet is improved and becomes robust to noises. Using visual backward masking as an example, we demonstrate that our model can explain visual cognitive behaviors that involve the interplay between two pathways. We hope that this study will provide insight into understanding visual information processing and inspire the development of new object recognition architectures in machine learning.

## 1 Introduction

Imagine you are driving a car on a highway and suddenly an object appears in your visual field, crossing the road. Your initial reaction is to slam on the brakes even before recognizing the object. This highlights a core difference between human vision and current machine learning strategies for object recognition. In machine learning, visual object recognition is often viewed as a feedforward, bottom up process, where object features are extracted from local to global in a hierarchical manner; whereas in human vision, we can capture the gist of a visual object at a glance without processing the details of it, a crucial ability for us (especially animals) to survive in competitive natural environments. This strategic difference has been demonstrated by a large volume of experimental data. For examples, Sugase et al. (1999) found that neurons in the inferior temporal cortex (IT) of macaque monkeys convey the coarse information of an object much faster than the fine information of it; FMRI and MEG studies on humans showed that the activation of orbitofrontal cortex (OFC) precedes that of the temporal cortex when a blurred object was shown to the subject (Bar et al., 2006); Liu et al. (2017) further demonstrated that the dorsal pathway extracts the coarse information of an object in less than 100ms after the stimulus onset, and this coarse information guides the subsequent local information processing.

Indeed, the Reverse Hierarchy Theory for visual perception has proposed that although the representation of image features along the ventral pathway goes from local to global, our perception of an object goes inversely from global to local (Hochstein & Ahissar, 2002). How does this happen in the brain? Experimental studies have revealed that there exist two anatomically and functionally separated signal pathways for visual information processing (see Fig.1). One is called the parvocellular

Figure 1: Illustration of the two separated pathways for information processing in the visual system. An image of an eagle is processed through two pathways. Upper panel: the P-pathway processes the detailed information of the image. Lower panel: the M-pathway processes the coarse information of the image rapidly, generates predictions about the image (association), and modulates the information processing of the P-pathway (feedback). MRGC: midget retina ganglion cell. PRGC: parasol retina ganglion cells. EVA: early visual area. LOC: lateral occipital complex. IPS: intraparietal sulcus. SC: superior colliculus. PFC: prefrontal cortex.

pathway (P-pathway), which starts from midget retina ganglion cells (MRGCs), projects to layers 3-6 in the lateral geniculate nucleus (LGN), and then primarily goes downstream along the ventral stream. The other is called the magnocellular pathway (M-pathway), which starts from parasol retina ganglion cells (PRGCs), projects to layers 1-2 of LGN, and then goes along the dorsal stream or the subcortical pathway (the superior colliculus and downstream areas). The two pathways have different neural response characteristics and complementary computational roles. Experimental findings have shown that the P-pathway is sensitive to colors and responds primarily to visual inputs of high spatial frequency; whereas the M-pathway is color blind and responds primarily to visual inputs of low spatial frequency (Derrington & Lennie, 1984). It has been suggested that the M-pathway serves as a short-cut to extract coarse information of images rapidly, while the P-pathway extracts fine features of images slowly, and the interplay between two pathways endows the neural system with the capacity of processing visual information rapidly, adaptively, and robustly (Bar, 2003; Wang et al., 2020; Bullier, 2001; Liu et al., 2017). For instance, by extracting the coarse information of an image, the M-pathway can generate predictions about what are expected in the visual field, and this knowledge subsequently modulate the fine information processing in the P-pathway (Fig. 1).

Although the existence of separated P- and M- pathways is well known in the neuroscience field, exactly how they cooperate with each other to facilitate information processing remains poorly understood. In this study, we build up a two-pathway model to elucidate the computational properties associated with the interplay between two pathways (Fig. 2). We use convolution neural networks (CNNs) as the building blocks, since recent studies have revealed that CNNs are effective to model the neuronal response variability along the visual pathway (Yamins et al., 2013; Kriegeskorte, 2015). Specifically, we model the P-pathway using a relatively deep CNN, which has small-size kernels and receives detailed visual inputs, referred to as FineNet hereafter. The M-pathway is modeled by a relatively shallow CNN, which has large-size kernels and receives blurred visual inputs, referred to as CoarseNet hereafter. Based on the proposed model, we investigate several computational issues associated with the interplay between two pathways, including how CoarseNet learns from FineNet via imitation, and how FineNet benefits from CoarseNet via feedback to leverage its performance. We also use the two-pathway model to reproduce the backward masking phenomenon observed in human psychophysic experiments.

## 2 THE TWO-PATHWAY MODEL

The structure of our two-pathway model is illustrated in Fig. 2, where FineNet and CoarseNet mimic the P- and M- pathways, respectively. Notably, FineNet is deeper than CoarseNet, reflecting that the P-pathway goes through more feature analyzing relays (e.g., V1-V2-V4-IT along the ventral pathway) than the M-pathway. FineNet also has smaller convolutional kernels than CoarseNet, reflecting that MRGCs in the retina have much smaller receptive fields than PRGCs. Furthermore, we consider that FineNet receives detailed and colourful visual inputs, reflecting that MRGCs have small receptive fields and are color sensitive; while CoarseNet receives blurred and gray inputs, reflecting that PRGCs have large receptive fields and are color blind.

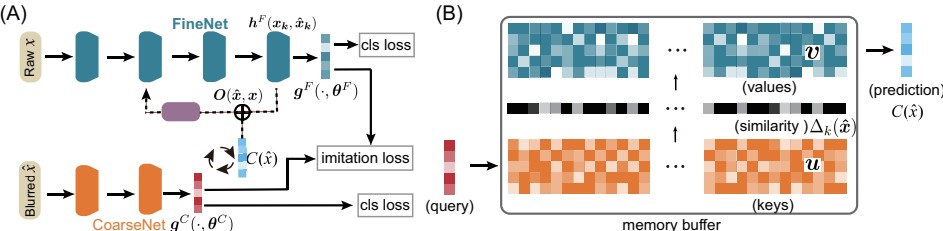

(A)    (B)

Figure 2: Illustration of the two-pathway model. (A) The architecture of the model. The blue and orange blocks represent the feedforward convolutional layers in FineNet and CoarseNet, respectively. The purple one represents the feedback convolution block in FineNet (for details, see Sec. 3.1). In inference, CoarseNet extracts coarse features $g^C(\hat{x})$ from a blurred image $\hat{x}$, which serves as a cue to predict the fine features $C(\hat{x})$ of the image via association. The associated result is then combined with the deep representations $h^F(x, \hat{x})$ to form a feedback signal $O(x, \hat{x})$, and the latter modulates an early layer of FineNet. In training, FineNet is optimized by minimizing the classification loss, and CoarseNet by minimizing both classification and imitation losses. (B) Illustration of static memory association (SMA). A query of the coarse features $g^C(\hat{x})$ of the input $\hat{x}$ is associated with a weighted summation of the fine features stored in the memory buffer, where the weighting coefficient $\Delta_k(\hat{x})$ is the similarity between the coarse features and the key vector $u_k$.

In the model, we consider that the two pathways interact with each other in three forms: 1) **Imitation learning**. Since CoarseNet has a shallow structure and receives blurred inputs, it is hard to train CoarseNet well for object recognition directly. Hence we consider that CoarseNet learns the feature representations of FineNet via an imitation process. Later we will argue that this has an important biological implication (Sec. 3.2). 2) **Association**. It is supposed that the M-pathway generates predictions about what might be in the visual scene, which guides the information processing in the P-pathway. We model this by considering that CoarseNet predicts the representation of FineNet through a memory association process. 3) **Feedback**. It is known that coarse information can serve as a cognitive bias guiding the extraction of fine information of images. We model this by feeding the associated prediction back to an earlier layer of FineNet to enhance the fine feature extraction. The details of the two-pathway model are introduced below.

## 2.1 THE INFERENCE PROCESS OF THE MODEL

Denote the input to FineNet as $x$ and the input to CoarseNet as $\hat{x}$. $\hat{x}$ is obtained by either filtering $x$ with a 2D Gaussian filter or binarizing $x$. Denote the output of CoarseNet to be $p^C(\hat{x}) = f^C\left[g^C\left(\hat{x}; \theta^C\right); w^C\right]$, where $g^C(\cdot; \theta^C)$ and $f^C(\cdot; w^C)$ represent, respectively, the feature extractor and the linear classifier of CoarseNet, and $\{\theta^C, w^C\}$ the trainable parameters. The output of FineNet is similarly denoted as $p^F(x) = f^F\left\{g^F\left[x, O(\hat{x}, x); \theta^F\right]; w^F\right\}$, where the feature extractor $g^F(\cdot; \theta^F)$ has an extra input component $O(\hat{x}; x)$, representing the feedback signal.

To generate the feedback signal $O(\hat{x}, x)$ in FineNet, we consider a memory association process. Two types of associations are exploited in this work, static memory association (SMA) and dynamic memory association (DMA) . They have the similar effect of using coarse features $g^C(\cdot; \theta^C)$ as a cue to predict fine features. SMA is simpler, but we also consider DMA, as it introduces temporal dynamics into our two-pathway model necessary for reproducing the backward masking experiment (see Sec. 3.6). For clearance, we only introduce SMA here (see Fig. 2B), and DMA is described in Appendix F. Specifically, we implement SMA with the cache memory model (Orhan, 2018), which performs a key-value association. The model stores a pair of a key matrix $u \in R^{d \times K}$ and a value matrix $v \in R^{c \times K}$ in the memory buffer, with $K$ the number of memory items and $d$, $c$ the dimensions of the key and value vectors, respectively. The columns $u_k$ and $v_k$ represent, respectively, the normalized $g^C(\hat{x}_k; \theta^C)$ of CoarseNet and the flattened feature vector $h^F(x_k, \hat{x}_k)$ of the last convolution layer of FineNet. When a specific query vector $g^C(\hat{x})$ of CoarseNet is presented, we first calculate its similarities with all key vectors stored in the memory buffer, which are given by $\Delta_k(\hat{x}) = \exp\left[\beta g^C(\hat{x})^\top u_k\right]$, for $k = 1, \dots K$, with $\beta$ controlling the sharpness of similarity. After that, we calculate the associated result, i.e., the predicted fine features, which is

given by $C(\hat{x}) = \sum_k v_k \Delta_k(\hat{x}) / [\sum_k \Delta_k(\hat{x})]$. The inference of FineNet forms a continuous loop so that the feedback signal is updated iteratively (see Fig. 2A). At time step $t$, the feedback signal in FineNet is calculated by $O_t(\hat{x}, x) = C(\hat{x}) + h_{t-1}^F(x, \hat{x})$. Notably, at the first step $t = 1$, only the associated result from CoarseNet is available, which gives $O_1(\hat{x}, x) = C(\hat{x})$. This reflects the fact that the M-pathway is much faster than the P-pathway, which generates the first feedback signal without interacting with high visual areas in the P-pathway.

In summary, the inference of the model involves interaction between two pathways: in response to an image, CoarseNet first generates its output and meanwhile predicts the fine features of FineNet through association; the predicted result is then combined with the deep representations of FineNet to form a feedback signal, which modulates the shallow layer of FineNet for feature extraction; this feedback loop can go on iteratively to continuously leverage the performance of FineNet.

## 2.2 THE TRAINING OF THE MODEL

During training, FineNet and CoarseNet are optimized jointly. To get the network output for an input, we run the feedback loop iteratively in FineNet for $T$ steps ($T = 2$ used in this study). FineNet is optimized through minimizing the cross-entropy loss, which is given by

$$L_F = -\frac{1}{N} \sum_{i=1}^{N} \sum_{j=1}^{K} y_{i,j} \ln p_j^F(x_i), \tag{1}$$

where $p_j^F$ is the $j$th element of $p^F$, i.e., the likelihood of the $j$th class, and $y_{i,j}$ is the $j$th element of the one-hot label $y_i$ for the image $x_i$, which is 1 for the correct class and 0 otherwise. The summation runs over all images $N$ and all classes $K$.

Since CoarseNet receives coarse inputs and has a shallow structure, we optimize it via a combination of classification and imitation losses, which is written as

$$L_C = \frac{1}{N} \sum_{i=1}^{N} \left[ -\alpha \sum_{j=1}^{K} y_{i,j} \ln p_j^C(\hat{x}_i) + \frac{1-\alpha}{2} \|g^C(\hat{x}_i) - g^F(x_i, \hat{x}_i)\|^2 \right], \tag{2}$$

where the symbol $\| \cdot \|$ denotes $L_2$ normal, and $\alpha$ is a hyper-parameter balancing the cross-entropy loss and the imitation loss.

Since SMA aims to store the long-term correlation (association) between the feature representations of CoarseNet and FineNet, we update its key and value matrices after every two training epochs.

## 3 INTERPLAY BETWEEN TWO PATHWAYS

### 3.1 IMPLEMENTATION DETAILS

Based on the proposed model, we carry out simulation experiments to explore the computational properties associated with the interplay between two pathways. Three datasets, Pascalvoc-mask, CIFAR-10, and CIFAR-100 are used (see details Appendix A). To generate blurred inputs $\hat{x}$ to CoarseNet, we either low-pass filter $x$ using a 2D Gaussian filter with $std = 2$ or binarizing $x$ using a shape mask (see examples in Fig. 7A and the details described in Appendix A.2). In the experiments, FineNet consists of four convolutional layers (Fig. 2A), each of which comprises a $3 \times 3$ convolution, followed by a group normalization, a ReLU nonlinearity. The numbers of convolutional filters in 4 layers are $[64, 128, 256, 512]$. CoarseNet consists of two convolutional layers with the same composition as in FineNet, except that it comprises 128-filter $7 \times 7$ convolution kernels in the first layer and 512-filter $5 \times 5$ convolution kernels in the second layer. Both FineNet and CoarseNet have a global pooling layer before the readout layer (for generating $g^C(\hat{x}; \theta^C)$ and $g^F(x, O(\hat{x}, x); \theta^F)$, respectively). The feedback kernel consists of an upsample layer and an $1 \times 1$ convolutional layer, followed by group normalization and sigmoid nonlinearity. It takes $O(\hat{x}, x)$ as the input and output a weighting term to modulate the representations in the second convolutional layer of FineNet via element-wise multiplication. The balancing term $\alpha$ is $0.4$ in Eq. 2. During training, the memory buffer in SMA is updated after every two epochs, and $\beta = 100$. Both FineNet and CoarseNet share the same training settings: the total number of training epochs is 150, SGD with a momentum term

0.9 is used to optimize parameters, and the initial learning rate is 0.05 which is multiplied with 0.1 after 100 and 125 epochs. For the training of dynamic memory association (DMA) implemented by RBM, please refer to Appendix F.1.

## 3.2 IMITATION LEARNING IMPROVES COARSENET

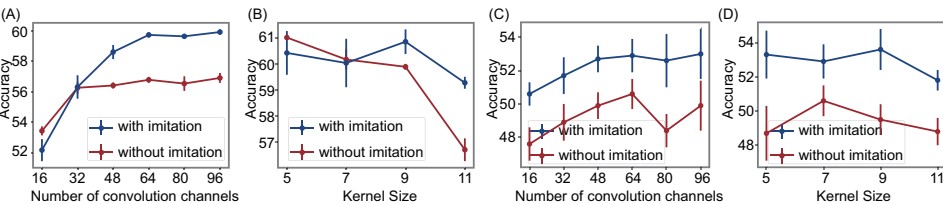

Figure 3: Imitation learning from FineNet improves the performance of CoarseNet. (A-B): performances of CoarseNet trained on low-pass filtered images from CIFAR10. (A) Performances vs. the number of convolution channels. (B) Performances vs. the size of convolution kernel. (C-D): performances of CoarseNet trained on the binarized Pascalvoc-mask. (C) Performance vs. the number of convolution channels. (D) Performance vs. the size of convolution kernel. See Appendix A for the details of the training and testing data.

In the two-pathway model, CoarseNet is supposed to have a good initial guess of the image, which serves as a cognitive bias to facilitate the performance of FineNet. However, since CoarseNet is shallow, has large convolution kernels, and receives coarse inputs, it is hard to train CoarseNet well independently. Therefore, we consider that CoarseNet learns the feature representations of FineNet via imitation learning. This is an important property, which may have some far-reaching implications to brain functions. We therefore carry out a separate experiment to study the effect of imitation learning. In the experiment, we focus on exploring whether CoarseNet can learn from FineNet via imitation, without considering other interactions between two pathways (the details of the experiment are described in Appendix B).

The results are presented in Fig. 3, which demonstrates that with imitation, the classification accuracy of CoarseNet is improved considerably, compared to that without imitation over a wide range of parameters. Specifically, with respect to the number of convolution kernels in CoarseNet, the improvement is significant when the number of kernels is large (Fig. 3A for low-pass filtered inputs; Fig. 3C for binarized inputs); with respect to the size of kernels in CoarseNet, the improvement is also significant (Fig. 3B for low-pass filtered inputs; Fig. 3D for binarized inputs). The fact that the effect of imitation learning also depends on the network parameters (Fig. 3A) indicates that in reality there is a trade-off between having a simple structure for the M-pathway and the capability of the M-pathway learning from the P-pathway.

From the computational point of view, the brain faces a difficulty of "designing" the M-pathway. On one hand, the M-pathway needs to be shallow and process coarse visual inputs in order to generate quick responses (which is important in a dangerous environment); on the other hand, the M-pathway needs to efficiently generate approximated, if not accurate, recognition of an object, serving as a good initial guess for further processing. However, it is a well-known fact that a shallow neural network alone is unable to achieve good object recognition (this has actually motivated the development of deep neural networks). So, how does the brain resolve this dilemma? Here, our study suggests that the strategy of imitation learning proposed in machine learning (Hinton et al., 2015) may provide a solution to this challenge, that is, the shallow M-pathway learns the representations of the deep P-pathway through imitation to improve its performance. Imitation learning may also be involved in other brain functions, such as for knowledge transfer and memory consolidation between hippocampus and neocortex (Alvarez & Squire, 1994; Sirota et al., 2003). During the acquisition of motor skills, it has been observed that neural activities gradually shift from the prefrontal cortex to the premotor, posteriorparietal, and cerebellar areas (the so-called scaffolding-storage proposed by (Petersen et al., 1998)), indicating that imitation learning may occur across cortical regions. Notably, the brain also has resources to implement imitation learning, e.g., the widely observed synchronized oscillations between cortical regions (Buzsáki & Draguhn, 2004) can modify neuron connections via

Hebbian plasticity to support the transfer of neural representations. It will be interesting to explore how imitation learning is realized in real neural systems.

### 3.3 TWO-PATHWAY PROCESSING IMPROVES ROBUSTNESS TO NOISE

| Models | Clean | Gaussian noise | Shot noise | Impulse noise | Adversarial noise |
|---|---|---|---|---|---|
| FineNet-only ($n_{fd} = 0$) | $86.9_{\pm 0.1}$ | $50.0_{\pm 0.5}$ | $57.8_{\pm 0.8}$ | $59.0_{\pm 0.4}$ | $-$ |
| FineNet-only ($n_{fd} = 1$) | $88.4_{\pm 0.0}$ | $56.6_{\pm 1.2}$ | $64.4_{\pm 1.2}$ | $61.4_{\pm 1.4}$ | $-$ |
| FineNet-only ($n_{fd} = 3$) | $88.0_{\pm 0.0}$ | $59.1_{\pm 1.4}$ | $65.9_{\pm 1.2}$ | $63.0_{\pm 0.1}$ | $-$ |
| two-pathway-FFL | $88.2_{\pm 0.2}$ | $58.8_{\pm 0.0}$ | $65.8_{\pm 0.2}$ | $62.7_{\pm 0.6}$ | $61.6_{\pm 0.1}$ |
| two-pathway-SFL | $\mathbf{88.6}_{\pm 0.1}$ | $56.3_{\pm 1.8}$ | $63.0_{\pm 1.4}$ | $63.2_{\pm 0.9}$ | $55.4_{\pm 0.2}$ |
| Our model | $86.7_{\pm 0.2}$ | $\mathbf{62.2}_{\pm 0.2}$ | $\mathbf{68.0}_{\pm 0.1}$ | $\mathbf{66.5}_{\pm 0.5}$ | $\mathbf{65.7}_{\pm 0.5}$ |

Table 1: The two-pathway model outperforms others on diverse noise perturbations. FineNet-only with $n_{fd} = 0, 1, 3$, refer to FineNet without feedback connection, with 1 loop of feedback interaction, and with 3 loops of feedback interaction, respectively. The models of two-pathway-FFL and two-pathway-SFL refer to two different ways of fusing the features of FineNet and CoarseNet. FineNet takes clean RGB images and CoarseNets the grayed, low-pass filtered images. Adversarial noises are generated by attacking FineNet using the Fast Gradient Sign Method (Goodfellow et al., 2014). The network performances for Gaussian, shot, and impulse noises are obtained by averaging over 5 different noise perturbation levels, and the results for adavesarial noises are obtained by averaging over 8 different noise perturbation levels. Mean and std are obtained by averaging over 4 trials. The noise generation details are described in Appendix A.2. Additional experimental results are presented in Appendix E.

A deep CNN trained for image classification is known to overly rely on local textures rather than the global shape of objects (Baker et al., 2018; Geirhos et al., 2018a;b), which is sensitive to unseen noises. In our model, since CoarseNet processes blurred visual inputs, whereby the local texture information is no longer the main cue supporting object classification, we expect that CoarseNet is robust to noise corruptions. Furthermore, through association and feedback, we expect that the robustness of FineNet to noises is also leveraged. We carry out simulations to test this hypothesis.

We compare our model with a single-passway model formed only by FineNet, referred to as FineNet-only. For fair comparison, we also include feedback loops between higher and lower layers in FineNet-only. We train the models on clean CIFAR10 dataset and test them by adding various noise perturbations, including Gaussian, shot, impulse and adversarial noises. Tab. 1 shows that compared to the single-pathway model, although the accuracy of our model on clean data is decreased a little bit, its robustness to noises is improved significantly, confirming that CoarseNet does contribute to improving the noise robustness of the model.

We also test different ways of integrating FineNet and CoarseNet. Precious works have studied the integration between two parallel networks (Hou et al., 2017; Zhu et al., 2018; Kim et al., 2019). However, they considered that the two networks share the similar structures and receive the same inputs, which are conceptually different from the brain-inspired two-pathway model. We borrow their methods of fusing networks to integrate FineNet and CoarseNet (while keeping all other hyper-parameters unchanged). Two fusion strategies, a simple fusion learning (SFL) and a feature fusion learning(FFL), are adopted (Kim et al., 2019). Both of them consider integrating information only at the top layers of two networks (for details, see Appendix C). As shown in Tab. 1, our model still has a much better performance on noise robustness than other methods, indicating that the association and feedback module in our model is suitable for integrating information between two pathways.

Remarkably, compared to other methods, our model does not have increased performances on classifying clean images, rather the performances are often decreased a little bit compared to that of a well-trained deep neural network. The reliable achievement of our model is the improved robustness to noises. This highlights an important goal for the brain employing two-pathway processing: the P-pathway learns to recognize objects with high fidelity based on fine features that are only available when images are clean; when images are ambiguous, the M-pathway relying on coarse features of objects compensates for noises. Certainly, there are other computational advantages associated with the two-pathway processing (see below).

### 3.4 ROUGH-TO-FINE PROCESSING IN THE TWO-PASSWAY MODEL

| Models | Clean | Gaussian noise | Shot noise | Impulse noise |
|---|---|---|---|---|
| FineNet-only ($n_{fd} = 1$) | $63.5_{\pm 0.3}$ | $28.2_{\pm 1.3}$ | $34.9_{\pm 1.2}$ | $28.6_{\pm 1.2}$ |
| Our model | $62.5_{\pm 0.3}$ | $\mathbf{30.5}_{\pm 0.7}$ | $\mathbf{36.6}_{\pm 0.7}$ | $\mathbf{31.0}_{\pm 0.4}$ |

Table 2: CoarseNet facilitates the performance of FineNet. The images are taken from CIFAR-100, which form 20 super-classes and 100 sub-classes (for details, see Appendix A). CoarseNet and FineNet perform super- and sub- class classifications, respectively. The experiment settings the same as in Tab. 1

In the above, we have considered that the two pathways process the same categorical information of images. In reality, the two pathways may process different levels of categorical information of images, and object recognition goes from rough to fine, for instance, CoarseNet may recognize the higher category of an object (e.g., animal), and FineNet recognizes the lower category of the object (e.g., cat). In such a case, the result of CoarseNet can still serve as a cognitive bias to facilitate the performance of FineNet. We carry out experiments to confirm this.

We construct a classification task, in which CoarseNet and FineNet are trained to recognize the super- and sub- classes of images, respectively, and the interaction between two networks remains unchanged as before. The results are presented in Tab. 2, which show that indeed the classification results on super-classes of images by CoarseNet can serve as a cognitive bias to improve the performance of FineNet on classifying the sub-classes of the images.

### 3.5 COMPONENT ANALYSIS OF THE TWO-PATHWAY MODEL

We further analyze the contributions of different elements in the model, and confirm that when any one of four elements of the model (FineNet, CoarseNet, the association module (SAM), and the feedback loop) is missing or changed, the robustness of the model to noise is degraded dramatically (see Fig. 4). Notably, these elements are also the key characteristics of the visual signal pathways.

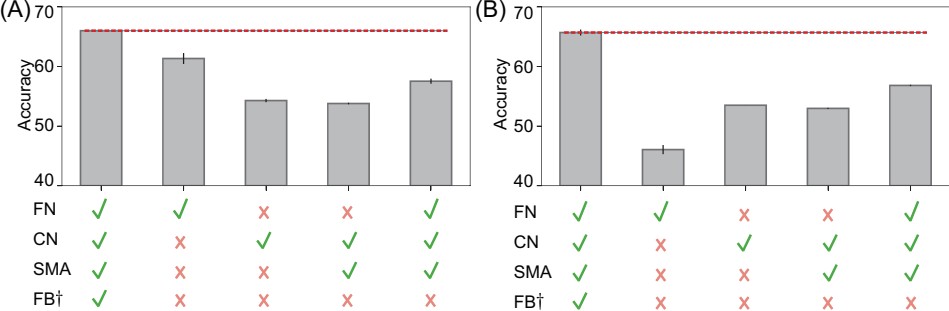

Figure 4: Component analysis of the two-pathway model. FN: FineNet. CN: CoarseNet. SAM: static associative memory. FB†: the long-range feedback from CoarseNet and the higher layer of FineNet to the second layer of FineNet; without FB†means a short-range feedback to the third layer of FineNet. Red dashed line: the performance of the two-pathway model. (A) Model performances with respect to Gaussian, impulse and shot noises. (B) Model performances with respect to adversarial noise. Experimental details are the same as in Tab. 1.

### 3.6 TWO-PATHWAY PROCESSING ACCOUNTS FOR VISUAL BACKWARD MASKING

Visual backward masking is a classic experiment widely used in cognitive psychology to investigate attention, awareness and dyslexia. In the cognitive experiment, a masking stimulus is presented after the target stimulus with a brief delay (usually 30-70 ms), which incurs a failure of the subject to consciously perceive the target (Fig. 5 A). Breitmeyer & Ganz (1976) proposed a psychological two-channel framework, a fast transient channel and a slow sustained channel, to explain the backward

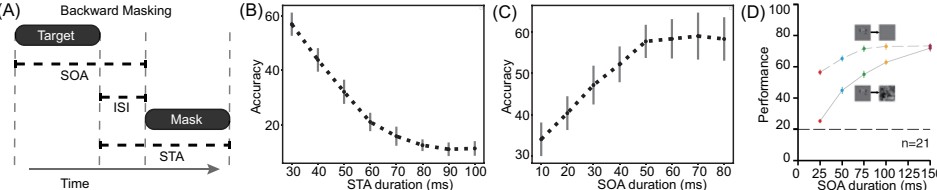

Figure 5: The two-pathway model reproduces the visual backward masking phenomenon. (A) A paradigm of the backward masking experiment, adapted from Macknik & Martinez-Conde (2007). SOA: stimulus onset asynchrony, the time interval between the onsets of target and mask; ISI: inter-stimulus interval, the interval between the termination of target and the onset of mask; STA: stimulus termination asynchrony, the interval between the terminations of target and mask. ISI=0 is used in our simulation. (B) Model performance vs. STA. (C) Model performance vs. SOA. (D) The experimental result, adapted from Tang et al. (2018).

masking phenomenon. The idea is that since the time delay for masking is too short, the neural representation of the mask, which is extracted rapidly through the transient channel, intercepts the neural representation of the target sustained in the slow channel, and hence disturb the perception. The important factors affecting the masking effect are the target and the mask durations, referred to as stimulus onset asynchrony (SOA) and stimulus termination asynchrony (STA), respectively.

Our two-pathway provides a natural computational model to explain the backward masking phenomenon. To capture the temporal effect, we consider a dynamical memory association process implemented by restrict Boltzmann machine (RBM), which holds the same idea of using coarse features as a cue to predict fine features as in SMA. The visible part of RBM is composed of the concatenated features from CoarseNet and FineNet, which are associated with each other through hidden variables (for the details, see Appendix F.1 and Appendix F.2). In the simulation, one iteration in RBM equals to a time step of 10ms. As suggested by the neural data (Sugase et al., 1999; Bar et al., 2006; Liu et al., 2017), information processing in the M-pathway proceeds that in the P-pathway for about 50 ms. Therefore, the coarse features are loaded into RBM 50 ms before their corresponding fine features. Because of this time lag, the coarse information of the mask is confounded with the fine information of the target, leading to wrong association that interferes the perception. The larger the STA or the shorter the SOA, the stronger interference of the mask. Our model successfully reproduces the backward masking effect as observed in the experiment. As shown in Fig. 5, the classification accuracy of the model increases with SOA, agreeing with the experimental findings in Tang et al. (2018).

## 4 CONCLUSION

In the present study, we have proposed a two-pathway model mimicking visual information processing in the brain. The model is composed of FineNet and CoarseNet, with the former extracting fine information of visual inputs and the latter extracting coarse information of inputs. CoarseNet processes information rapidly, whose result serves as a feedback to facilitate the performance of FineNet. Our study reveals several appealing properties associated with the interplay between two pathways, which are: 1) through imitation, CoarseNet can learn from FineNet to improve its performance; 2) through association and feedback from CoarseNet, the robustness to noise of FineNet is improved significantly; 3) the result of CoarseNet can serve as a cognitive bias to leverage the performance of FineNet, achieving rough-to-fine information processing; 4) the two-pathway model can explain the visual backward masking phenomenon as observed in the experiment. To our knowledge, our work is the first one that builds up a network model to mimic the structures of the two biological visual pathways and studies their interaction properties. We hope that this study will give us insights into understanding visual information processing and inspire the development of new object recognition architectures in machine learning.

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

# A DATASETS AND MANIPULATION OF THE INPUT

## A.1 THREE DATASETS

We use three datasets, Pascalvoc-mask, CIFAR-10 and CIFAR-100 to evaluate our model. Pascalvoc-mask and CIFAR-10 are used to demonstrate the effect of imitation learning in Fig. 3. CIFAR-10 and CIFAR-100 are used to test the noise robustness and the rough-to-fine processing property of our model.

**Pascalvoc-mask** is a new dataset we created from the Pascalvoc2012 dataset (Everingham et al., 2015), which contains 20 foreground object classes. The goal of the original Pascalvoc2012 dataset is to recognize objects from a number of visual object classes in realistic scenes. There are two main tasks (classification and detection) and two additional competitions (segmentation and action classification). In the current study, we are only interested in those objects with precise annotated segments. Totally, there are 2913 images with 6866 objects having annotated segments. To create the Pascal-mask dataset, firstly, we extract each object image from the raw image and each object segment from the corresponding "SegmentationObject" image set according to the bounding box information (see Fig. 6A). Secondly, we remove object images with low resolution (the number of pixels in width or height is less than 50) or large aspect ratio (width/height or height/width is more than 3). In this way, the number of remaining objects is 4887. Thirdly, to obtain the masked counterpart of each object, we gray the object segment by setting the pixel values for objects to be 1 and backgrounds to be 0 (see Fig. 6B). All object images are resized to $64 \times 64$ to fit the input of CoarseNet. The new dataset consists of 4887 object images with masks. We split the dataset into 4512 training and 375 testing images, where the testing set is all from the Pascalvoc2007 testing set. See Tab. 3 for the details of Pascalvoc-mask. The dataset can be found at `https://drive.google.com/file/d/1TP0QsFBtVwXaCENGTwuk9ZhDlkGMyTOj/view?usp=sharing`.

| aeroplane | bicycle | bird | boat | bottle | bus | car | cat | chair | cow |
|---|---|---|---|---|---|---|---|---|---|
| 104/10 | 147/15 | 215/11 | 126/9 | 92/6 | 174/12 | 206/16 | 261/17 | 398/33 | 204/10 |
| diningtable | dog | horse | motorbike | person | pottedplant | sheep | sofa | train | tvmonitor |
| 115/14 | 267/16 | 176/11 | 162/18 | 1017/100 | 186/14 | 188/21 | 178/14 | 138/12 | 158/16 |

Table 3: The number of samples in each class of Pascalvoc-mask. Digits in each column mean the training/testing numbers. The number of classes is 20 and the total number of training examples/testing examples is 4512/375.

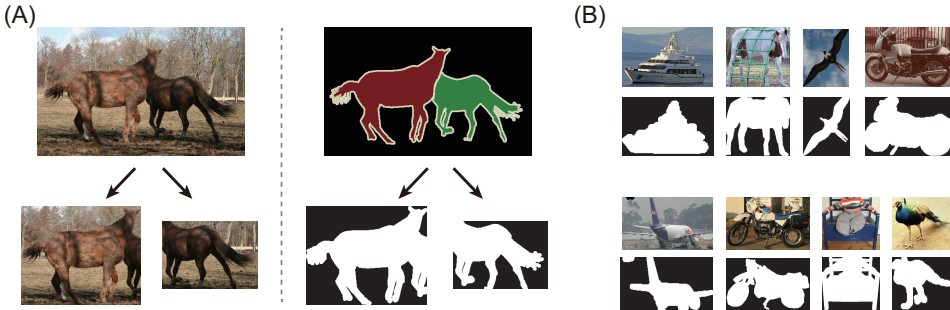

Figure 6: Data examples of the dataset Pascalvoc-mask. (A) left panel: a raw image and the corresponding objects; right panel: the segment of the raw image and the corresponding object segments. (B) Examples of Pascalvoc-mask. Images in the RGB channel and their masked counterparts.

**CIFAR-10** consists of 60000 $32 \times 32$ colour images for 10 classes, with 6000 images per class, and they are split into 5000 training and 1000 test images in each class.

**CIFAR-100** is a harder version of the CIFAR-10 dataset, which has 100 classes, with 600 images per class, and they are split into 500 training and 100 testing images in each class. The 100 classes in

the CIFAR-100 are further grouped into 20 superclasses, so that each image has a pair of sub-class and super-class labels (this information is used to test the rough-to-fine processing property).

## A.2 MANIPULATING THE INPUT WITH DIFFERENT NOISES

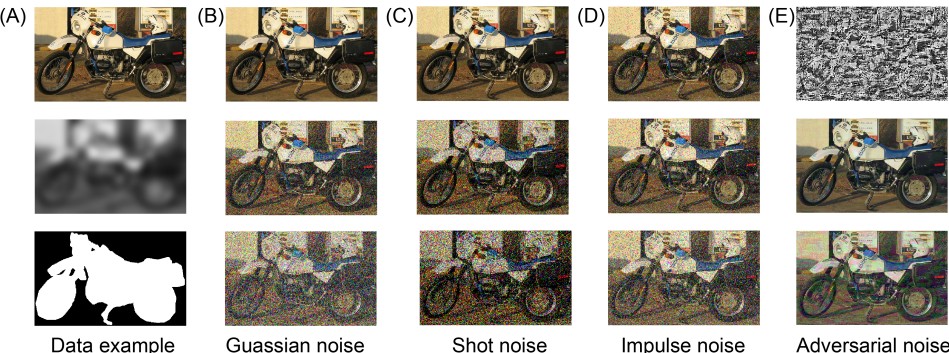

Figure 7: Examples of noise disruptions used in the experiments. (A) Examples of visual inputs used for training FineNet and CoarseNet. From up to down, a raw image to FineNet, the corresponding low-pass filtered image (blurred) to CoarseNet, and the corresponding binarized image (mask data) to CoarseNet. (B-E) Different kinds of noise disruptions. (B) Examples of Gaussian noise with $std = 0.04, 0.3, 0.6$, respectively. (C) Examples of shot noise with $c = 100, 3, 1$, respectively. (D) Examples of impulse noise with $p = 0.07, 0.15, 0.3$, respectively. (E) Adversarial noise. Up: the adversarial noise of the example image in (A)-up, obtained by the Fast Gradient Sign Method (Goodfellow et al., 2014); Middle and down: the adversarial examples with the noise levels of $0.1$ and $0.5$, respectively.

Evaluating the model performances under different kinds of noise disruption is a main task in the current study. Here we describe the details of manipulating inputs with various forms of noise. Four types of noises are used, Gaussian, shot, impulse, and adversarial noises. Please see Fig. 7 for details.

We obtain the performances of models on Gaussion noise, shot noise, and impulse noise dataset by averaging over 5 amplitude levels (Fig. 4). For Gaussian noise, the 5 levels correspond to the noise variance $std = [0.04, 0.06, 0.08, 0.09, 0.10]$. For shot noise, the 5 levels correspond to the multiplication parameter $c = [500, 250, 100, 75, 50]$. For impulse noise, the 5 levels correspond to the probability $p = [0.01, 0.02, 0.03, 0.05, 0.07]$. For adversarial noise, we average 9 different levels with $\epsilon = [0.005, 0.01, 0.015, 0.02, 0.025, 0.03, 0.035, 0.04]$.

## B IMPLEMENTATION DETAILS OF IMITATION LEARNING

In Sec. 3.2, we illustrate the effect of imitation learning to CoarseNet. Without loss of generality, FineNet and CoarseNet both adopt simpler structures than that used in the noise robustness task. FineNet used in this task consists of three stacked layers, each of which comprises a 128-filter $3 \times 3$ convolution, followed by a batch normalization, a ReLU nonlinearity, and $2 \times 2$ max-pooling. CoarseNet has two stacked layers with the same composition as in FineNet, except that it comprises 64-filter $11 \times 11$ convolution in the first layer and 128-filter $9 \times 9$ convolution in the second layer. The balancing term $\alpha = 0.4$ is used when training CoarseNet. Both FineNet and CoarseNet have a fully-connected layer of 1000 units before the readout layer. Except for normalizing with the channel-wise mean and standard deviation of the whole dataset, no other pre-processing strategies are adopted. All other settings are the same as described in Sec. 3.1.

## C   IMPLEMENTATION DETAILS OF SFL AND FFL

In Sec. 3.3, we mentioned two feature fusion methods SFL and FFL. Here we describe the details. In machine learning society, many feature fusion methods have been proposed to fuse the features from parallel networks. However, these methods focused on fusing features from networks which take the exact same input and share similar model structures, while our two-pathway model takes different inputs and structures. We concatenate the features of top convolutional layers in FineNet and CoarseNet, and then perform the fusion operation through a convolutional structure (Kim et al., 2019).

SFL and FFL share the same feature fusion module which consist of a depthwise convolution and a pointwise convolution. They are different in training objective functions. In SFL, the sub-network classifiers and the fusion classifier are both trained with cross-entropy losses simultaneously. In FFL, in additional to cross-entropy losses, networks are trained via mutual knowledge distillation losses, which was proved to be a more effective fusion method (Kim et al., 2019). Note that both FFL and SFL have more trainable parameters than our model (see Tab. 4).

Table 4: The number of trainable parameters in different models. FineNet has more trainable parameters than CoarseNet. Networks with FFL and SFL have more trainable parameters than our two-pathway model.

| Models | Number of Parameters |
|---|---|
| FineNet | $3,119,808$ |
| CoarseNet | $827,456.$ |
| FFL | $4,410,112$ |
| SFL | $4,410,112$ |
| two-pathway | $3,942,144$ |

## D   EFFECTS OF $\beta$ IN THE SAM BUFFER

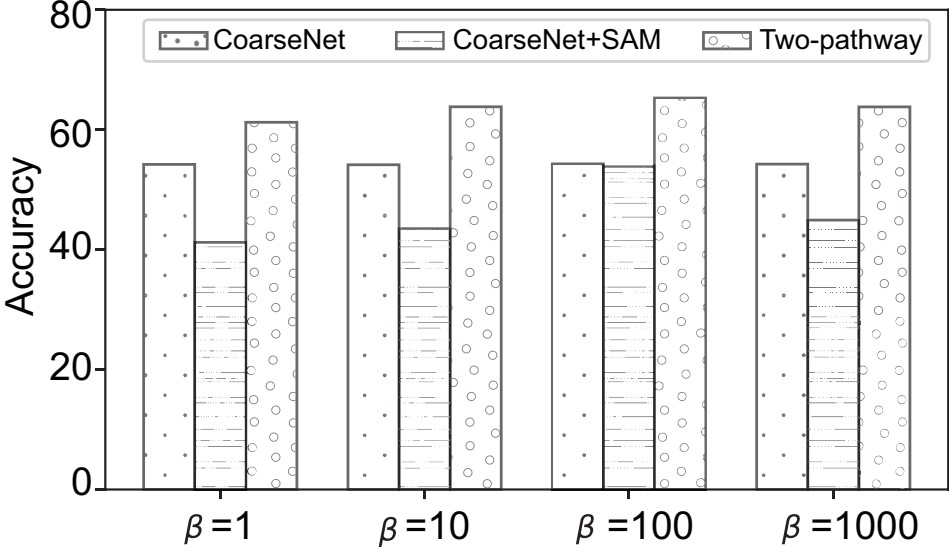

Figure 8: Performance of the two-pathway model against the parameter $\beta$ in SMA. CoarseNet and the two-pathway model are trained under different $\beta$ settings. When $\beta = 100$, both the two-pathway model and CoarseNet which reads out via SAM achieve the best performances.

In practice, the value of $\beta$ affects the performance of our two-pathway model. We find that a good $\beta$ for CoarseNet, which makes a prediction via SAM (Orhan, 2018), is always a good one for the

corresponding two-pathway model ( See the three bars when $\beta = 100$ in Fig. 8). So the value of $\beta$ can be chosen by training a CoarseNet alone, and then we find a good $\beta$ for CoarseNet through parameter searching.

# E   ADDITIONAL EXPERIMENTAL RESULTS ON THE NOISE ROBUSTNESS EVALUATION TASK

| Models | Clean | Gaussian noise | Shot noise | Impulse noise | Adversarial noise |
|---|---|---|---|---|---|
| $std = 1$ | | | | | |
| FFL | $88.2_{\pm 0.2}$ | $63.1_{\pm 2.1}$ | $\mathbf{69.6_{\pm 1.6}}$ | $66.0_{\pm 0.7}$ | $62.8_{\pm 0.5}$ |
| SFL | $\mathbf{88.6_{\pm 0.1}}$ | $55.6_{\pm 1.0}$ | $62.2_{\pm 0.3}$ | $63.4_{\pm 0.9}$ | $55.2_{\pm 0.3}$ |
| two-pathway | $86.7_{\pm 0.2}$ | $\mathbf{64.1_{\pm 1.0}}$ | $69.5_{\pm 1.1}$ | $\mathbf{67.3_{\pm 0.4}}$ | $\mathbf{65.6_{\pm 0.2}}$ |
| $std = 2$ | | | | | |
| FFL | $88.2_{\pm 0.2}$ | $58.8_{\pm 0.0}$ | $65.8_{\pm 0.2}$ | $62.7_{\pm 0.6}$ | $61.6_{\pm 0.1}$ |
| SFL | $\mathbf{88.6_{\pm 0.1}}$ | $56.3_{\pm 1.8}$ | $63.0_{\pm 1.4}$ | $63.2_{\pm 0.9}$ | $55.4_{\pm 0.2}$ |
| two-pathway | $86.7_{\pm 0.2}$ | $\mathbf{62.2_{\pm 0.2}}$ | $\mathbf{68.0_{\pm 0.1}}$ | $\mathbf{66.5_{\pm 0.5}}$ | $\mathbf{65.7_{\pm 0.5}}$ |
| $std = 3$ | | | | | |
| FFL | $88.2_{\pm 0.2}$ | $58.3_{\pm 3.7}$ | $65.4_{\pm 2.6}$ | $63.5_{\pm 2.0}$ | $60.2_{\pm 0.4}$ |
| SFL | $\mathbf{88.3_{\pm 0.1}}$ | $56.5_{\pm 0.1}$ | $63.5_{\pm 0.4}$ | $62.0_{\pm 0.2}$ | $56.4_{\pm 0.4}$ |
| two-pathway | $87.5_{\pm 0.1}$ | $\mathbf{60.0_{\pm 1.5}}$ | $\mathbf{66.4_{\pm 1.0}}$ | $\mathbf{65.3_{\pm 0.1}}$ | $\mathbf{65.8_{\pm 0.2}}$ |
| FFL* | $89.4_{\pm 0.3}$ | $54.4_{\pm 0.2}$ | $62.6_{\pm 0.4}$ | $62.2_{\pm 0.1}$ | $67.7_{\pm 0.2}$ |

Table 5: Performances of different models under different noise levels. CoarseNet in these models takes grayed and low-pass filtered inputs with $std = 0, 1, 2, 3$. FineNet takes clean inputs. Different from FFL, FFL* consists of two FineNets. Mean and std are obtained by averaging over 4 trials.

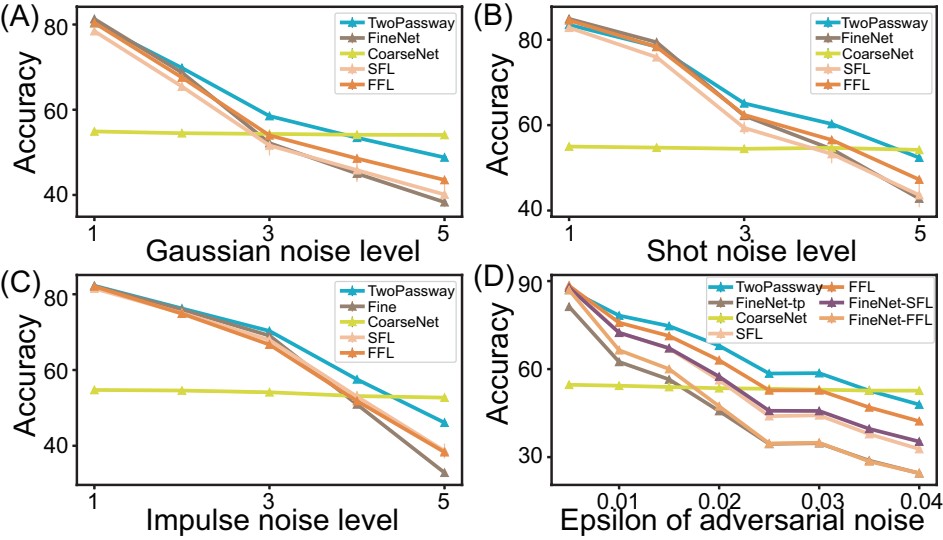

Figure 9: Model performances against noises and adversarial noise perturbations. In (A-C), model performances against noise perturbations. FineNet and CoarseNet are trained independently on the dataset. (D) Model performances against adversarial noise perturbations. FineNet-tp,FineNet-FFL and FineNet-FFL means FineNet in the two-pathway model, FFL, and SFL, respectively. Noise perturbation details are shown in Appendix A.2. Experimental details are the same as that in Tab. 1.

We also run additional experimentt conditions. As shown in Tab. 5, our model outperforms other methods significantly in various noise tasks and with Gaussian filters of diffrent $std$.

As shown in Fig. 9, although CoarseNet has much lower accuracy compared to FineNet, it is very robust to all kinds of noise disruption and meanwhile its performance decreases much slower than

other models with the noise disruption level. Then robust CoarseNet can generate a robust cue to SAM to associate a relatively clean predictive representation, and the predictive presentation can help to improve the performance of FineNet dramatically.

## F  IMPLEMENTING BACKWARD MASKING WITH OUR TWO-PATHWAY MODEL

We introduce the details of modeling the backward masking phenomenon using our two-pathway model. First, we introduce RBM used as the association module. Then we elucidate how the two-pathway model works on this task. Last, we introduce some experimental findings on the backward masking task.

### F.1  RESTRICT BOLTZMANN MACHINE (RBM) AS A DYNAMICAL MEMORY ASSOCIATION MODEL

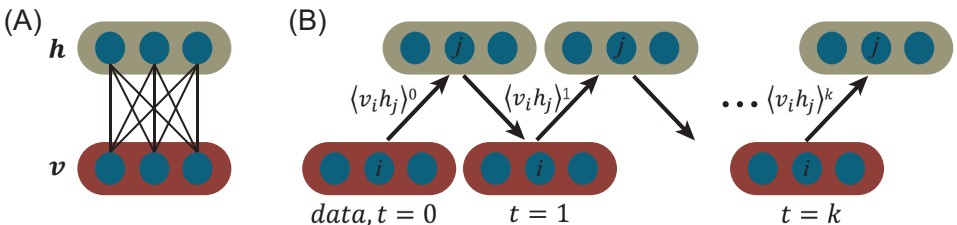

Figure 10: Learning in RBM. (A) Diagram of a RBM with three visible and three hidden units. There are only connections between layers. (B) Illustrating the Gibbs sampling process in RBM during training. At time $t = 0$, the visible units $v$ are initialized and the hidden units are updated according to $h \sim P(h|v)$. At time $t = 1$, the visible units are updated according to $v \sim P(v|h)$ and the correlations $< v_i h_j >$ are the statistics used for contrastive learning in the RBM. The number of units in the visible layer is 1000, with 500 units for coarse features and 500 units for fine features. For simplicity, we only test our model on the Mnist dataset.

To investigate the effect of different factors on the target visibility, e.g., the task duration (SOA), the mask duration (STA), we modified the similarity-based association phase into RBM, which introduces dynamics in the association phase. RBM is a simplified version of Boltzmann Machine (BM), with the latter being an extension of the Hopfield model with stochastic dynamics (Hinton et al., 2006). Both BM (Ackley et al., 1985) and the Hopfield model (Hopfield, 1982) can be used to capture how memory patterns are stored as stationary states of neural circuits via recurrent connections between neurons. RBM consists of a visible and a hidden layers with no within-layer connections. Denote the input to the visible layer as $v$, activities at the hidden layer as $h$ and the connection matrix between two layers is $W$. The energy function of a RBM is written as

$$E(v, h) = -av^T - bh^T - vWh^T, \tag{3}$$

where $a$ and $b$ represent the bias vectors in the visible and the hidden layers, respectively. The joint probability of a configuration $(v, h)$ is written as

$$P(v, h) = \frac{e^{-E(v, h)}}{Z}, \tag{4}$$

where $Z$ is the partition function given by $Z = \sum_h \sum_v e^{-E(v, h)}$. The probability of a specific $v$ is

$$P(v) = \frac{1}{Z} \sum_h e^{-E(v, h)}. \tag{5}$$

The appealing property of the bipartite graph structure of RBM is that the conditional distributions $P(\boldsymbol{h}|\boldsymbol{v})$ and $P(\boldsymbol{v}|\boldsymbol{h})$ are factorial, i.e.,

$$P(\boldsymbol{v}|\boldsymbol{h}) = \prod_{i}^{n_v} P(v_i|\boldsymbol{h}), \quad P(v_i = 1|\boldsymbol{h}) = \sigma(a_i + \sum_{j=1}^{n_h} w_{ij}h_j), \tag{6}$$

$$P(\boldsymbol{h}|\boldsymbol{v}) = \prod_{i}^{n_h} P(h_i|\boldsymbol{v}), \quad P(h_i = 1|\boldsymbol{v}) = \sigma(b_i + \sum_{j=1}^{n_v} w_{ji}v_j), \tag{7}$$

where $\sigma(x) = 1/(1 + e^{-x/T})$ is a sigmoid function, with $T$ the temperature.

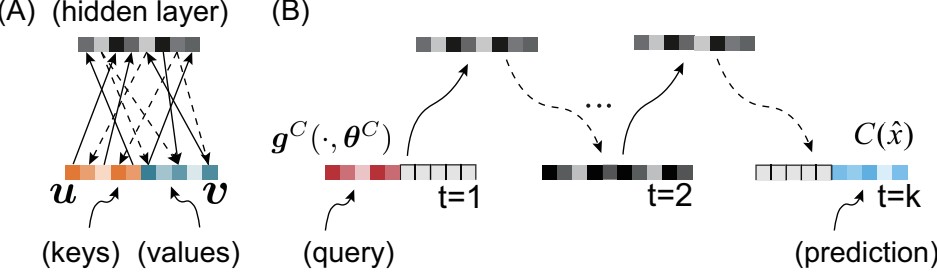

Figure 11: DMA implemented by RBM. (A) The training phase (see Fig. 10 for the details). (B) The retrivela phase. The coarse probe $g^C(\hat{\boldsymbol{x}})$ of a visual object is fed into the visible layer of the trained RBM and the prediction $\boldsymbol{O}(\hat{\boldsymbol{x}})$ is retrieved through the dynamics of RBM.

To implement the association phase, we construct $\boldsymbol{v}$ by concatenating the features from both CoarseNet and FineNet, e.g., $\boldsymbol{v} = [\boldsymbol{v}^C, \boldsymbol{v}^F]$, with $\boldsymbol{v^C} = g^C(\hat{x})$ and $\boldsymbol{v^F} = g^F(x)$. Given the training examples (features of input images), RBM is optimized through minimizing the negative log-likelihood:

$$L_{RBM} = -\frac{1}{N}\sum_{i=1}^{N} \log\left[P(\boldsymbol{v}_i)\right] = -\frac{1}{N}\sum_{i=1}^{N} \log\left[P(\boldsymbol{v}_i^F, \boldsymbol{v}_i^C)\right]. \tag{8}$$

The derivative of the log likelihood with respect to a connection weight is calculated to be,

$$\frac{-\partial \log P(\boldsymbol{v})}{\partial W_{ij}} = <v_i h_j>_{data} - <v_i h_j>_{model}, \tag{9}$$

where the first and the second terms in the right hand of the equation denote expectations over the distributions of data and the model, respectively. The first expectation is tractable. For the second expectation, we apply the strategy of contrastive divergence (CD) gradient (Hinton, 2002), which approximates the expectation over the model distribution by a sample generated via a number of Gibbs sampling iterations, with the initial state of the visible units being the training sample, as illustrated in Fig 10B. More specifically, we use the correlation statistics $<v_i h_j>^k$ after $k$ step Gibbs sampling to replace the $<v_i h_j>_{model}$ to update the connection weights, i.e.,

$$\Delta W_{ij} = \epsilon(<v_i h_j>^0 - <v_i h_j>^k), \tag{10}$$

where $\epsilon$ is the learning rate. During the training, $\boldsymbol{v}$ and $\boldsymbol{h}$ are sampled from $P(\boldsymbol{h}|\boldsymbol{v})$ and $P(\boldsymbol{v}|\boldsymbol{h})$ alternatively. The total number of training epochs is 2000. We use SGD to optimize the RBM with an initial learning rate of 0.1, which is multiplied with 0.1 after 500 and 1000 epochs.

Once the training is finished, we can feed a partial feature to the visible layer of RBM, and retrieve the complete one. For example, given a partial feature $\boldsymbol{v_0}$ at time 0, the hidden representations of RBM is $\boldsymbol{h_0} = P(\boldsymbol{h} = 1|\boldsymbol{v_0} = \sigma(\boldsymbol{a} + \boldsymbol{W}\boldsymbol{v_0})$ and the updated activation in the visible layer is $\boldsymbol{v_1} = \sigma(\boldsymbol{b} + \boldsymbol{W^T}\boldsymbol{h_1})$. After $k$ iterations, we can get a $\boldsymbol{v_k}$ which is a complete feature corresponding to $\boldsymbol{v_0}$ (see Fig. 11 B).

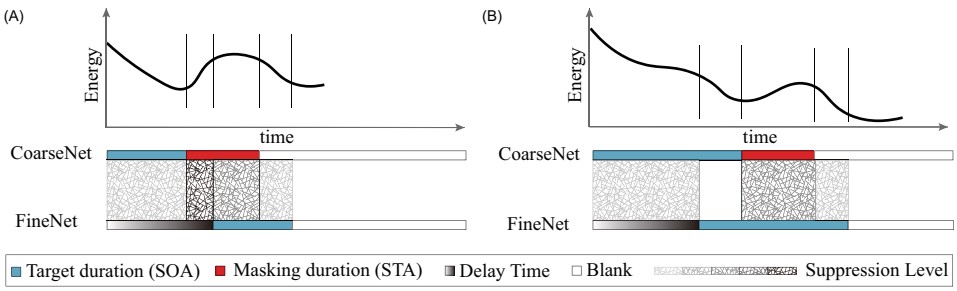

Figure 12: Illustration of the backward masking with our two-pathway model. (A) The effect of a small target duration to the network (small SOA). (B) The effect of a large target duration to the network (large SOA).

## F.2 INPUT TO RBM IN THE TASK

In Sec. 3.6, we showed that the fast representation of the mask processed along CoarseNet will suppress the slow representation of the target processed along the FineNet, which decreases the target visibility. The dynamics in RBM enables us to investigate which factors, such as the task duration (SOA) and the mask duration (STA), will affect the target visibility. Due to slow processing by FineNet, information arrived at RBM from the fine pathway has a lag of about 50 ms (biologically) with respect to that from the coarse pathway (see Fig. 12). If we set the processing time of each iteration to be 10 ms, then RBM will only receive coarse features $v^C$ as the input at the first 5 iterations. When the target information from FineNet arrives at the visible layer of RBM, it will interact with the features from CoarseNet (note that at this moment the features can be the mask or the coarse feature of the target, depending on the SOA value). The network evolves for a relatively long time (500 ms) and the final value of the fine part in the visible layer is used for recognition. Fig. 12A (B) shows the effect of a small (large) SOA value to the network performance. Energy is measured as in Eq. 3, with the lower the energy, the better the network performance (the better target visibility). Specifically, with a small target duration, such as SOA=30, which is smaller than the lag between two pathways, the network state is strongly interfered by the mask (red part in Fig. 12 A), and disrupts the target information from Finenet. When SOA is larger than the lag, the network state is only partially disrupted by the mask from CoarseNet, leading to a better performance. This result has been reported in Fig. 5D.

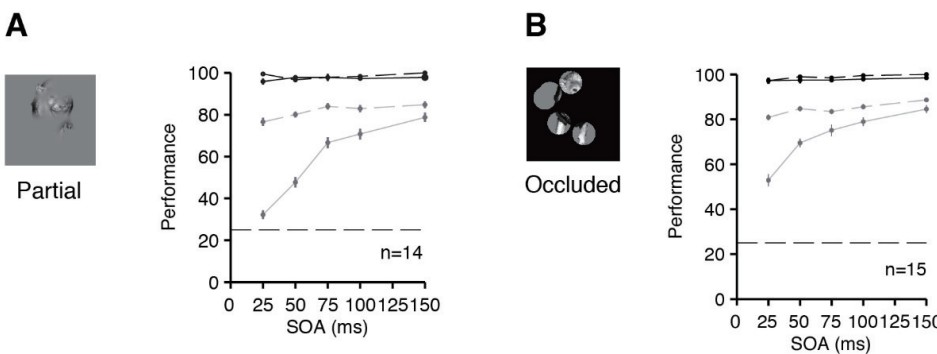

Figure 13: Performances of human subjects in the backward masking task. The figure is adapted from Tang et al. (2018). (A) Categorization performance on the partial stimuli on a set of 16 exemplars belonging to 4 categories (chance = 25%, the dashed line). 14 subjects participated in this task. (B) Occluded stimuli was used. Another 15 subjects participated in this task. The black lines indicate whole objects and the gray lines indicate the partial and occluded objects. Solid lines represents performance with masking and dashed lines without masking.

### F.3 Backward masking in psychophysical experiments

Here we introduce some results of backward masking in psychophysical experiments. They are from Tang et al. (2018), where subjects performed a 5-classes recognition task involving categorization of objects that were either partially visible or fully visible. Images were followed by either a gray screen (without masking) or a spatially overlapping noise pattern (with masking). SOA varies from 25 to 150 ms in randomly ordered trials (for the detialed setting, please see the original paper). Fig 13 shows that: 1) for whole objects, subjects perfectly recognized the objects, no matter what kind of disruption was used. Even with masking, the performance made no difference with that without masking. We obtain similar results as in Fig. 5 B&C. In Fig. 5B, we show that with a low mask noise level (can be viewed as non-masking), our model achieves a high performance. Fig. 5C shows that when STA is small (also can be viewed as non-masking), our model achieves a high performance; 2) For disrupted images, subjects' performances were gradually improved when the target duration increased from 25ms to 150 ms (the solid gray lines in both panels). This is also reflected in Fig. 5D in our model. Note that Tang et al. (2018) also proposed a attractor-based recurrent model to explain their experiment findings. They proposed that local recurrent connections could explain the backward masking phenomenon, which is different from our two-pathway model.

