# OpenReview forum: "Vision at A Glance: Interplay between Fine and Coarse Information Processing Pathways"
_ICLR.cc/2021/Conference — Reject_

### Official Review · AnonReviewer1 · 2020-10-28
**Interesting and important topic but poor technical soundness and contribution**

**Rating:** 3
**Confidence:** 4

**Review:**

**Summary**
This paper presents a new type of brain-inspired dual-pathway DNN model where the coarse (faster, less accurate) and fine (slower, more accurate) visual pathways augment each other during training and inference (via imitation and feedback) to boost the network's robustness to various noises.

**Pros**
(1) The topic is interesting and important as the P and M pathways are both crucial elements of the efficient and robust human visual system but their computational models are much less studied.
(2) The proposed model is new and overall simple to implement.

**Cons**
(1) The model's design decisions are arbitrary and poorly justified.
(Imitation learning) It's unclear why CoarseNet activations must mimic FineNet activations, since only FineNet activations are ultimately used for inference (not CoarseNet activations, which are further transformed and used by the FineNet). To justify the necessity of imitation learning, the authors should present full FineNet+CoarseNet results (not just Fig 3) using all 3 losses versus using only FineNet's classification loss.
(Binary masks) The setting that the CoarseNet can use clean binary masks of objects as inputs doesn't make sense at all in my opinion. Segmentation itself is a complex task that often requires high-level vision so it's unclear to me why the authors assume that PRGCs can provide such information.
(SMA) It's unclear why the SMA's u and v are implemented as memory buffers (that are updated per 2 epochs, which seem arbitrary and not biologically plausible either) following [Orhan, 2018], instead of end-to-end learned parameters (that are consistently updated every iteration with the rest of the network using backprop gradients). More generally, the authors should present solid reasons why association should also produce results that mimic FineNet activations.
(DMA and feedback) It's unclear why an RBM is required as an additional component to introduce dynamics, since extending the feedback loop dynamically (i.e. more loops) should also be able to achieve similar results using existing components (which seems more biologically plausible in terms of resources).

(2) The experimental results are weak and incomplete, and comparisons against related work are missing.
(Noise robustness) As the key benefit claimed in this paper, the noise robustness of the proposed model is however weak (still roughly 20% accuracy drops for all types of noises) and only better than simplified or similar variants of the model. In addition, using only FGSM (targeting the FineNet) to generate adversarial noises doesn't fully test the robustness of the model, since more recent techniques [1, 2] can easily generate smooth adversarial examples that will likely severely affect the CoarseNet (unlike FGSM).
(Related work) Although the authors argued that existing models are conceptually different, it doesn't mean that architecturally similar models [3, Hou et al.], SOTA in adversarial defense [4, 5], and most importantly other brain-inspired models, targeting robustness [6, 7] or not [8, Tang et al.], shouldn't be compared against to properly prove the value of this work. Also, as the field has started to more directly use neural data to guide better network design [8], it's unclear why the authors seem to have completely omitted this approach.
(Rough-to-fine processing) The value of this approach is unclear since the accuracies of training both networks using the subclass labels (CIFAR-100) are missing.
(Backward masking) Visual results alone (Fig 5 and 12) don't properly support the claim that this model "can explain visual cognitive behaviors that involve the interplay between two pathways". Please consider providing more detailed statistical analyses (e.g. R^2) if the authors want to make this claim.

(3) The clarity needs improvement.
The clarity of this paper is substandard as many key details are ambiguous or completely missing. For example, how does the SMA memory buffer store features? Random sampling over the training set? Is the model also trained on noisy data? How does a CN+SMA model (Fig 4) even work? What exactly are the backward masking stimuli used in the experiments (frame by frame)?

**Recommendation**
I recommend rejection of the paper given the following two major cons (see details above).
(1) The model's design decisions are arbitrary and poorly justified.
(2) The experimental results are weak and incomplete, and comparisons against related work are missing.

**Questions**
Please address the cons listed above.

**Additional Feedback**
(1) Although using an L2 loss for imitation learning is straightforward mathematically, the authors' arguments regarding how the brain may implement imitation learning aren't very convincing (Sec 3.2). For example, is there direct evidence of synchronized oscillation supporting the transfer of neural representations and thus imitation learning?
(2) Speed seems to be a major potential benefit of the proposed model, which however was not clearly discussed or benchmarked. Please consider adding speed comparisons against SOTA networks in terms of inference speed.

**References**
[1] Low Frequency Adversarial Perturbation, UAI, 2019
[2] SmoothFool: An Efficient Framework for Computing Smooth Adversarial Perturbations, WACV, 2020
[3] U-Net: Convolutional Networks for Biomedical Image Segmentation, MICCAI, 2015
[4] Feature Denoising for Improving Adversarial Robustness, CVPR, 2019
[5] Adversarial Defense by Restricting the Hidden Space of Deep Neural Networks, ICCV, 2019
[6] Brain-inspired Robust Vision using Convolutional Neural Networks with Feedback, NuerIPS-W, 2019
[7] Biologically Inspired Mechanisms for Adversarial Robustness, arXiv, 2020
[8] Brain-Like Object Recognition with High-Performing Shallow Recurrent ANNs, NeurIPS, 2019

---

> ### Author Response · Authors · 2020-11-24
> **Our model’s design decisions are not arbitrary but based on biological constraints conceptually.**
>
> Thanks for your careful review.
> (1)	1) Our model’s design decisions are not arbitrary but based on biological constraints conceptually. For example, FineNet and CoarseNet mimic P-pathway and M-pathway, that two pathways interacts with an associative memory has been proposed by moshe bar (moshe bar, Nature Neuroscience Review, 2002), feedback connections widely exists in our ventral visual pathway. 2) Imitation loss has little effect on the performance of our two pathway model (not shown in the paper), because of network structures and imitation method adopted here. What we want to illustrate here is that our brain do perform imitation learning (see Sec.3.2) and it may be a good solution for our brain to improve the network’s performance and get a better tradeoff between performance and coarse structure. 3) There are some RGC types that can detect global object motion which may provide the binary mask [1].  4) Recurrent neural network can be also used, here we mainly follow the hypothesis that two pathways can interact via an associative memory (moshe bar, Nature Neuroscience Review, 2002) .5) It is a good idea, two pathway model may also be able to achieve similar results using existing components without RBM. We will try later.
>
> (2)	It is difficult to compare our model with the methods you mentioned here. In our two-pathway model, CoarseNet takes coarse inputs and is robust to noise perturbations. CoarseNet can provide a robust cognitive bias or prior which can help modulate the FineNet’s earlier representations and constraint the networks’ decisions. So, the robustness of CoarseNet affects the FineNet’s performance a lot. Although we have illustrate that CoarseNet help FineNet is non-trivial, our CoarseNet here is still oversimplified. How to design the CoarseNet is still an open question here.
>
> (3)	SMA memory buffer stores features of training data. In our two-pathway model, FineNet is trained on clean data, CoarseNet is trained on coarse data which is low-pass filtered data here. In Fig4, SMA stores the pairs of features and one-hot label vectors, CoarseNet can do classification via association in SMA, which we want to illustrate that noise robustness in our model comes from the interplay between two pathways but not the associative process. The backward masking stimuli used in the experiments is frame by frame.
>
>
> [1]. Tim Gollisch, et al, Eye Smarter than Scientists Believed: Neural Computations in Circuits of the Retina, Neuron, 2010.

---

### Official Review · AnonReviewer3 · 2020-10-30
**Results are not strong, and descriptions are not clear enough**

**Rating:** 3
**Confidence:** 4

**Review:**

This paper proposed a two-pathway neural network to mimic the interplay between the parvocellular (slow and fine-grained) and magnocellular (fast and course) pathways in neural systems. The two pathways are named as FineNet and CourseNet. During inference, the FineNet received recurrent feedback signals from the CoarseNet via an attention layer and memory. During training, cross-entropy loss are used for both pathways, and an "imitation" loss is used to encourage the CoarseNet pathway to mimic the FineNet

First, a coarse-fine two-pathway network is not novel, as coarse-fine or multi-scale pathways is a well-known design used in computer vision applications. Using an attention module for the coarse-to-fine interaction recurrently might be new, but the paper does not show if it could outperform simpler interactions.

Second, the "imitation" loss is essentially model distillation. It is well-known that learn a weaker network by distilling a stronger network can result in stronger results for the weaker network.

Third, the accuracy on Cifar-10/100 are low, far away from the well-established SOTA. Though the paper provides a few ablations, observations made on a too9 weak model are not conclusive.

Finally, the paper might aim to be explanatory, but it falls short in clarity. The model description in Sec 2.1 is elusive and not sufficiently detailed. The choice of the number of feedback steps is ad hoc. The outreach to backward masking oversold a computational analogy as a neural computational model while providing a vague explanation of the background and results of the experiments.

---

> ### Author Response · Authors · 2020-11-24
> **We are not focusing on the SOTA results but aiming to proposed a new direction that has not yet been explored in machine learning.**
>
> Thanks for the reviewing. Below are our replies to your concerns.
>
> The reviewer has largely misunderstood the purpose of our work. We do not aim to add to existing machine learning models and make slight improvements, rather, we propose a new direction that has not yet been explored in machine learning. Problems such as visual stability (noise robustness), rough-to-fine processing, visual backward masking have been largely studied in the cognitive neuroscience community, but rarely studied in the machine learning community. Tackling these issues not only provides a way to explain the working mechanism of biological vision, but also makes up for the deficiency of current machine learning models.
>
> 1, Two branch models proposed in the computer vision applications are for dealing with specific problems such as action recognition and scene understanding while ours  is motivated by the multiple processing pathways in the biological visual system and focuses more on the cognitive aspects which are rarely studied in the machine learning society.
>
> 2, Yes, the "imitation loss" is indeed the knowledge distillation loss which aims to help the learning of CoarseNet while it takes only very coarse inputs and this is important for the association network to propose more accurate predictions from the coarse features.
>
> 3, SOTA results on CIFAR 10/100 can be achieved by only tuning the FineNet which is not the focus of the current model. The function of the CoarseNet is to provide a quick prediction of what the input might be so that the ambiguities in the visual inputs can be efficiently resolved. This can be reflected on the noise robustness, rough-to-fine processing and visual masking phenomenon, but can not be reflected on the clean input data which aims to improve the classification accuracy.
>
> 4,  We will adopt the suggestions by the reviewer and  make our paper more clear.

---

### Official Review · AnonReviewer4 · 2020-11-01
**a novel biologically inspired neural network**

**Rating:** 6
**Confidence:** 4

**Review:**

This paper proposes a dual-path CNN architecture with complementary roles (FineNet and CoarseNet) which is inspired by parvocellular and magnocellular pathways in the primate brain. The CoarseNet receives blurred inputs and has large kernels while FineNet received high-resolution input, is deep and has small kernels. It is shown that this architecture improves the robustness in object recognition performance over single-pathway architecture and could replicate the behavioral responses in humans during a classic psychological experiment (backward masking).

The proposed architecture is novel and the results support the main claims regarding the improvements in robust object recognition behavior and replication of the psychological experiment. My main criticism of this work is the lack of comparison with alternative models in the reported results. For example results in table 1 and 2 are only compared to alternative settings of the same model but no other studies or SOTA.

The authors could further improve the manuscript by considering the following comments.

* what is the neuroscience evidence of relative shallowness of M-pathway?
* The processes controlling the memory buffer has not been explained at all.
* The exact images used to produce each of the results are not clearly explained.
* In Table-1, it is unclear what "our model” is and how is it different from FFL and SFL?
* In sectino 3.3, in the FineNet-only models, it is unclear how the feedback loop is resolved in the absence of CoarseNet.
* In section 3.3 it is stated that “this highlights an important goal for the brain employing two-pathway processing”. While this is a plausible hypothesis about the role of these two pathways, we don't know if that is indeed the evolutionary goal of having two pathways. The dorsal pathway does much more than fast recognition of objects e.g. motion perception. I suggest the authors either remove or revise this statement.
* In tables 1-2, are the reported results the output of FineNet? It is not currently clear if that is the case
* page 8: restrict Boltzmann machine —> Restricted Boltzmann Machine
* For results in Fig-5, how would alternative models like CORnet [1] perform on this task?

[1] Kubilius, Jonas, et al. "Brain-like object recognition with high-performing shallow recurrent ANNs." Advances in Neural Information Processing Systems. 2019.

---

> ### Author Response · Authors · 2020-11-24
> **Our two-pathway model focuses on the cognitive aspects of deep neural networks (haven't been tackled before), so that the comparisons are very restricted.**
>
> We acknowledge the careful and valuable comments of the reviewer.
>
> In fact, there have been very little works modeling the interplay between different information processing pathways in the primate brain before. Hence, to our best kwoledge, we haven't found any related works which can be compared to.  In the machine learning society, many works are established and validated by comparing with previous related works. However, our focus is on how to explain some interesting and important aspects that current machine learning models have ignored but the primate brain is very good at. A compromise is to evaluate our model on the image classification benchmarks such as MSCOCO, ImageNet, etc. However this is not our main concern, as we believe that the single FineNet model with more stacked layers can handle the image classification problem very well. Tasks such as noise robustness, rough-to-fine processing and explaining the psychophysical phenomenon, i.e., the backward masking effect are much more attractive which haven't been considered by the machine learning community yet.
>
> Single neuron recordings (Sugase et al., 1999), MEG and FMRI studies (Bar et al., 2006; Liu et al, 2017) have shown that information processed on the M-pathway is faster than the P-pathway (about 50ms) which means that the CoarseNet takes fewer information processing hierarchies. Traditional findings have shown that the dorsal pathway is very important for motion perception, spatial perception, etc. But recent findings have shown that it is also very important for the quick acquisition of global gist in the visual scene (Liu et al, 2017).
>
> Thanks for the advice of the reviewer, we will explain all these concerns in the future version of our paper.

---

### Decision · Program_Chairs · 2021-01-07
**Final Decision**

**Decision:**

Reject

**Comment:**

This paper explores a network that has a parvo (fine, detailed, slow)
and magno (low-res, quick) stream.  The ideas are interesting and the
results intriguing, and one reviewer is in favor of acceptance.
Several reviewers criticized the clarity of the paper. and the lack of
details for, explanations of, and critical evaluation of, the design
decisions.  For example, how do the results depend on certain design
decisions?  I think that with a bit more work, this paper has potential to
be a very impactful paper.  I would encourage the authors to follow the
detailed suggestions and resubmit the work to a high-impact conference or
journal.